# QTL Mapping and Functional Identification of Candidate Genes Regulated by *Sinorhizobium fredii* HH103 and Associated with Nodulation Traits in Soybean

**Hejia Ni [†], Siyi Tian [†], Guoqing Zhang, Jingyi Huo, Huilin Tian, Yang Peng** 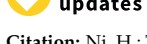 **, Kaixin Yu, Qingshan Chen, Jinhui Wang, Dawei Xin *** 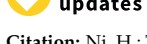 **and Chunyan Liu ***

Key Laboratory of Soybean Biology of the Chinese Ministry of Education, Key Laboratory of Soybean Biology and Breeding, Genetics of Chinese Agriculture Ministry, College of Agriculture, Northeast Agricultural University, Harbin 150036, China; nhjwinner@163.com (H.N.); tsynly@163.com (S.T.); 17745156274@163.com (G.Z.); 18545566677@163.com (J.H.); thl00002@163.com (H.T.); b210301014@neau.edu.cn (Y.P.); 18946018702@163.com (K.Y.); qshchen@neau.edu.cn (Q.C.); 18904818156@163.com (J.W.)
* Correspondence: xdawei@163.com (D.X.); cyliucn@163.com (C.L.)
† These authors contributed equally to this work.

**Abstract:** Large amounts of nitrogen fertilizer are annually applied to improve soybean yield. However, the overuse of nitrogen fertilizers has resulted in significant environmental pollution. Biological nitrogen fixation is an ecological and environmentally friendly method to increase soybean yield. However, the signaling pathway and function of genes in the plant host regulated by rhizobia under the symbiotic relationship remain unknown. In this study, the chromosome region in soybean responsive to *Sinorhizobium fredii* HH103 was identified using chromosome-segment-substituted lines produced from a cross between cultivated soybean SN14 and wild soybean Zyd00006. In addition, candidate genes associated with nodulation traits and regulated by *S. fredii* HH103 were identified. In total, three quantitative trait loci (QTLs) containing 68 genes were identified on chromosomes 02, 05, and 20. The differentially expressed genes among the QTL regions were determined using RNA-seq and qRT-PCR. *Glyma. 05G240500*, a potential gene responding to *S. fredii* HH103 and positively regulating soybean nodulation, was identified. To explore the relationships between haplotypes and soybean nodulation phenotypes, SNPs found in the regulatory areas of *Glyma.05G240500* haplotype were investigated. Our study revealed the role of *Glyma.05G240500* in symbiosis and provided a reference for facilitating symbiotic nitrogen fixation in the field and for marker-assisted selection.

**Keywords:** soybean; quantitative trait locus; symbiosis; nitrogen fixation; chromosome segment substitution lines

## 1. Introduction

Soybean (*Glycine max* [L.] Merr.) is one of the most economically important leguminous crops. Its worldwide production provides more than 25% protein for consumption by humans and animals [1,2]. Nitrogen (N) is necessary for soybean growth, yield, and quality. Annually, large amounts of nitrogen fertilizer are applied to improve soybean yields [3]. However, the overuse of nitrogen fertilizers has resulted in significant environmental pollution, deteriorating the quality of the soil, water resources, and even the atmosphere due to the fact that a small amount of nitrous oxide released during nitrification mixes into the stratosphere, where it destroys ozone [4–6]. Legumes produce flavonoids through their root exudates to attract rhizobia and generate root nodules. Nodules act as a biological nitrogen fixation factory for their plant hosts, effectively turning atmospheric $N_2$ into ammonia [7,8]. This ecological and environmentally friendly biological nitrogen fixation method can increase soybean yield and reduce the application of nitrogen fertilizer and production costs. It is crucial that biological nitrogen fixation (BNF) contribute to the store

of N that is available for the crop. The most significant N$_2$-fixing agents in farming systems are the symbiotic relationships between legumes and rhizobia [9]. For the application of such symbioses in agricultural development, it is crucial to understand the molecular mechanism of the symbiosis between legumes and rhizobia [8].

Many soybean-yield-related traits have been studied to increase soybean yield and balance soybean supply and demand [10,11]. The 100-seed weight (100-SW) is an important component of soybean yield traits and can characterize soybean yield to a certain extent [12]. A 11.5-Mb region on chromosome 10 in soybean was identified to be associated with both yield and seed weight [11]. Quantitatively inherited soybean-yield-related traits such as flowering and maturity are influenced by both internal and external factors [13]. In addition, isoflavones, proteins, and oil contents are the most important qualitative characteristics in soybeans and typical quantitative traits [14]. According to Jin et al., increasing nodule formation enhanced soybean utilization of N and P, which enhanced yield performance [15]. Symbiotic nitrogen fixation (SNF) in legumes significantly contributes to the improvement of crop quality and yield by supplying nitrogen resources. The SNF ability of leguminous plants is considered a complex quantitative trait, involving many related characteristics such as nodule number (NN), nodule dry weight (NDW), nodule fresh weight (NFW), etc. [16,17]. Total nitrogen derived from the atmosphere (Ndfa) refers to the amount of nitrogen fixed from the atmosphere through symbiotic nitrogen fixation. Three QTLs for percentage of nitrogen derived from the atmosphere (%Ndfa) and five QTLs for Ndfa were identified, respectively [18]. These identified QTLs could accelerate the development of soybean varieties with enhanced SNF through marker-assisted breeding. Interestingly, a single gene locus can also regulate many plant traits, which are crucial for SNF. The acetylene reduction activity (ARA) of each plant, the acetylene reduction activity per nodule weight (ARA/NW), the acetylene reduction activity per nodule number (ARA/NN), the NN of each plant, the NW of each plant, and the QTL of stem length were observed to be co-located on chromosome 4 [16].

Liu et al. (2021) confirmed that *Glyma.02G135100*, *Glyma.02G100800*, *Glyma.02G109100*, and *Glyma. 02G113800* are affected by NopT (a nodulation outer protein) in the QTL, which may influence the NN in soybean [19]. Wang et al. (2020) identified two candidate genes, *Glyma.19g068600* and *Glyma.19g069200*, through population QTL mapping and haplotype analysis, and the expression levels of these two candidate genes affected the *S. fredii* HH103 infection in soybean [20]. Ni et al. (2022) revealed QTLs related to nodulation traits in soybean using chromosome segment substitution line populations and located 17 genes that may interact with NopL and NopT. These putative genes may correspond to root-specific and nodule-specific co-expression subnetworks, according to gene annotations [21]. These candidate genes screened by quantitative trait mapping could be used in map-based gene cloning for genetic improvement in soybeans or marker-assisted selection in breeding.

To effectively identify advantageous genes from wild soybean, using the wild soybean Zyd00006 as the donor parent and soybean cultivar SN14 as the recurrent parent, a population of chromosomal segment substitution lines (CSSLs) was produced in 2016 [22]. In this study, the response of the CSSL population to *S. fredii* HH103 inoculation was assessed. Further, by combining QTL and RNA-seq studies, putative genes associated with SNF were identified. Additional genetic, transgenic, and haplotypic studies revealed that *Glyma.05G240500* may be associated with nodulation regulation. The information provided by our study on the function of *Glyma.05G240500* in nodulation can help in the efficient use of SNF in soybeans.

## 2. Materials and Methods

### 2.1. Strains, Plasmids, and Soybean Genetic Materials

In this study, the following strains were used: *Escherichia coli* DH5α, *S. fredii* HH103, *Agrobacterium tumefaciens* EHA105, and *A. rhizogenes* K599. The plasmids used for the construction of vectors are listed in Supplementary Table S1. *E. coli* strains were cultured in LB media supplemented with appropriate antibiotics at 37 °C. HH103 was cultured in TY

medium supplemented with rifampicin at 28 °C, in conjunction with other *Agrobacterium* strains, in the appropriate media with appropriate antibiotics. Rifampin, kanamycin, and spectinomycin were used at a working concentration of 50 g/mL. In this study, 195 lines from a wild soybean CSSL population that covered 82.6% of the wild soybean genome were used [22].

### 2.2. Nodulation Tests and Detection of Nitrogenase Activity

Soybean germplasms SN14 and Zyd00006 and the CSSL population were used for nodulation tests. Healthy seeds were placed side by side in Petri dishes and placed in a vacuum desiccator in a fume hood. The seeds were incubated for 16–20 h in a desiccator with chlorine generated in a 100-mL beaker containing 100 mL of 10% sodium hypochlorite and 4 mL of 12 mol·L$^{-1}$ HCl. Five seeds for each soybean variety were allowed to sprout in sterile plastic jars with vermiculite as the top layer and nitrogen-free nutritional solution as the bottom layer. The higher and lower layers were bound together using cotton sheets [21]. The seedlings were grown in a greenhouse with 16 h of light per day at 26 °C and 8 h of darkness per day at 25 °C [23]. Soybean seedlings were inoculated with *S. fredii* HH103 when they reached the Vc stage (cotyledon stage). NN and NDW were measured to assess nodulation [28 days after inoculation (dpi)]. The statistical significance of the difference between NN and NDW was determined using the student's two-tailed *t*-test via SPSS 12.0. Nitrogenase activity in the root nodules was measured using the Gremaud and Harper (1989) acetylene reduction technique [24].

### 2.3. Whole-Genome Sequencing of the CSSL Population and Natural Varieties of Soybean

Using the Illumina HiSeq 4000 sequencing platform and the Williams 82 genome as the reference genome, the genomes of SN14, Zyd00006, the CSSL population, and 685 soybean wild types were sequenced for the current study (unpublished data). The SNPs were examined as described previously. FastP was used to ensure that the raw data was of high quality. The quality filtering parameters applied using FastP were "-c-f10-F10" with version 0.23.1. After collecting clean data, Williams82 v2.0 was used as the reference genome, and the Burrows–Wheeler Alignment (BWA, Version: 0.7.17-r1188) tool was used for mapping [25]. Besides, SNPs were identified in the SN14 and Zyd00006 genotypes using the Genome Analysis Toolkit (GATK) [26].

### 2.4. QTL Mapping and Screening of Candidate Genes

For QTL identification, the nodule phenotype of CSSL inoculated with wild strain HH103 was used. Through composite interval mapping, WinQTL Cartographer was used to identify QTLs for nodulation-related traits [27]. Due to variations in chromosomal substituted regions and genetic information from wild soybean, the nodulation phenotypes, such as NN, NDW, and nitrogenase activity, of the CSSL population inoculated with HH103 were different. The genomic regions associated with the nodulation phenotype after inoculation with HH103 were identified via QTL analysis based on the nodulation tests on the CSSL lines. Based on the combined analysis of QTL mapping and chromosomal substituted segments, genes from the localized regions identified by QTL mapping that might be relevant to nodulation were verified for their involvement in nodulation in further investigations.

### 2.5. Verification of Candidate Genes Using qRT-PCR

After HH103 inoculation or control treatment, the patterns of candidate gene expression in SN14 and Zyd00006 were examined using qRT-PCR. In brief, the roots were harvested 24 h after HH103 inoculation, snap-frozen in liquid nitrogen, and pulverized to a fine powder in a precooled mortar with liquid nitrogen. Total RNA was extracted using the TRIzol reagent. The PrimeScript RT reagent Kit (Takara Biotech Co., Beijing, China) was used to synthesize cDNA. qRT-PCR was performed using SuperReal PreMix Color (SYBR Green) (Vazyme Biotech, Nanjing, China). The experiment involved three

technical replicates. As a normalization control, *GmActin* (*Glyma.19G147900*) was used. The expression patterns of potential candidate genes were evaluated using $2^{-\Delta\Delta Ct}$ method.

### 2.6. Nodulation and Hairy Root Transformation in Soybean

Hairy-root transformation was conducted using *A. rhizogenes* strain K599 that contains pSoy1-*Glyma.05G240500*-GFP, pSoy1-GFP, pB7GWIWG2- *Glyma.05G240500*-DsRed, and pB7GWIWG2-DsRed [28]. GFP and DsRED were used as markers to assess positive hairy roots using LUYOR3415RG. To determine the impact of gene overexpression and interference on nodulation, hairy roots were inoculated with *S. fredii* HH103. At 28 dpi, nodulation was evaluated using NN and NDW as the parameters. The t-test was performed to determine the significance of changes in NN and NDW using three biological replicates, each with 15 plants.

### 2.7. Subcellular Localization of Candidate Genes

Using soybean cultivar SN14, the coding regions of all candidate genes were amplified via RT-PCR. These genes were inserted into the pGWB5 vector, which has a CaMV35S promoter and a tag for green fluorescent protein (GFP). Supplementary Table S2 lists the gene-specific primers used to clone the putative candidate genes. A positive control (empty vector) or the fusion proteins were transiently expressed in tobacco leaves. The subcellular localization of GFP-tagged proteins was observed 24 h after polyethylene glycol (PEG)-mediated protoplast transfection (Yoo et al., 2007). The GFP-tagged proteins were identified and located using confocal laser-scanning microscopy (LSM 710, Carl Zeiss, Jena, Germany) with a 488 nm argon ion laser. The excitation/emission wavelengths for detecting GFP and chlorophyll autofluorescence were 488 nm/507 to 535 nm and 610 nm/650 to 750 nm, respectively.

### 2.8. Phylogenetic Analysis of the Candidate Gene

The full-length protein sequence of Glyma.05G240500 from *G. max*, GlysoPI483463.05G206500 from *G. soja*, Vigun03g010300 from *Vigna unguiculata*, Phvul.002G322700 from *Phaseolus vulgaris*, Lj4g0019304 from *Lotus japonicus*, Ca_10406 from *Cicer arietinum*, Tp57577_TGAC_v2_mRNA19939 from *Trifolium pratense*, Medtr8g105780 from *Medicago truncatula*, arahy.Tifrunner.gnm1.ann1.B3N TUN from *Arachis hypogaea*, Lalb_Chr16g0387451 from *Lupinus albus*, MD13G1018800 from *Malus domestica*, Caden.03G117800 from *Castanea dentata*, Prupe.1G335100 from *Prunus persica*, Potri.010G104300 from *Populus trichocarpa*, and Thecc.02G308200 from *Theobroma cacao* were downloaded and aligned to create a phylogenetic tree based on the neighbor-joining method with the default parameter value using MEGA 5.0 software [29,30].

### 2.9. Transcriptome Analysis

The SN14 plants were inoculated with *S. fredii* HH103 or MgSO$_4$ buffer (control). The roots of HH103-inoculated and control plants were harvested at 0, 12, 24, 48, and 72 h post-inoculation (hpi). Each group contained three biological replicates. cDNA libraries were constructed using total RNA collected using the Illumina TruSeq Kit (Illumina Inc., San Diego, CA, USA). The Biomarker Technologies (Beijing, China) Illumina HiSeq 4000 PE 150 platform was used to sequence the cDNA library. Raw RNA-Seq data were aligned with the GlymaWm82.a2.v1 genome using Hisat2 [31]. DESeq2 was used to generate and standardize the Fragments Per Kilobase of Transcript Sequence per Million Base Pairs Sequenced (FPKM) values. Additionally, using HTseq-count, the read counts for each gene were obtained. The threshold for the differentially expressed genes (DEGs) was set at an absolute value of log2(fold-change) $\geq$ 2 and a false discovery rate (FDR) $\leq$ 0.01 for comparisons with the appropriate control treatment at each time point [32].

### 2.10. Haplotype Analysis of Candidate Gene

*Glyma.05G240500* was the subject of a haplotype analysis in 685 wild soybean varieties and the CSSL population. The genomes of the 685 natural varieties of soybean and the

CSSL population were sequenced to generate the genomic sequence of *Glyma.05G240500*, containing the coding sequence and a 3.0-kb promoter sequence. The genomic sequences were subjected to local BLAST analysis to obtain SNP information [33].

## 3. Results

### *3.1. Improved Cultivar SN14 and Wild Soybean Zyd00006 Exhibited Different Phenotypes after HH103 Infection*

We determined the phenotypes of SN14 and Zyd0006 after HH103 inoculation to identify the variation in nodulation due to HH103. HH103 inoculation led to a considerable difference in NN and NDW between SN14 and Zyd006. In terms of root nodule quantity, NDW, and nitrogenase activity 4 weeks after inoculation, SN14 and Zyd00006 significantly outperformed uninoculated plant material (Figure 1A–C). To further confirm the promotion of symbiotic nodulation due to HH103 inoculation, we examined the expression patterns of nodulation-related marker genes, namely, *GmNIN*, *GmNSP1*, and *GmNSP2*, in the inoculated soybean roots at 48 hpi and 72 hpi, and all four genes were observed to be induced by HH103 (Figure 1D). All 195 CSSLs exhibited substantial variation in nodulation pattern after HH103 infection, with NN ranging from 3 to 267 and NDW ranging from 0.0017 to 0.218. Within these ranges, the parental phenotypes were identified (Table 1). The genetic background of the CSSL population is more complex than that of the individual parents, SN14 and Zyd00006. Because the inherited features of individuals greatly varied, their responses to HH103 varied. The CSSL population with a more complicated genetic background should facilitate the mining of QTLs associated with the nodulation phenotype.

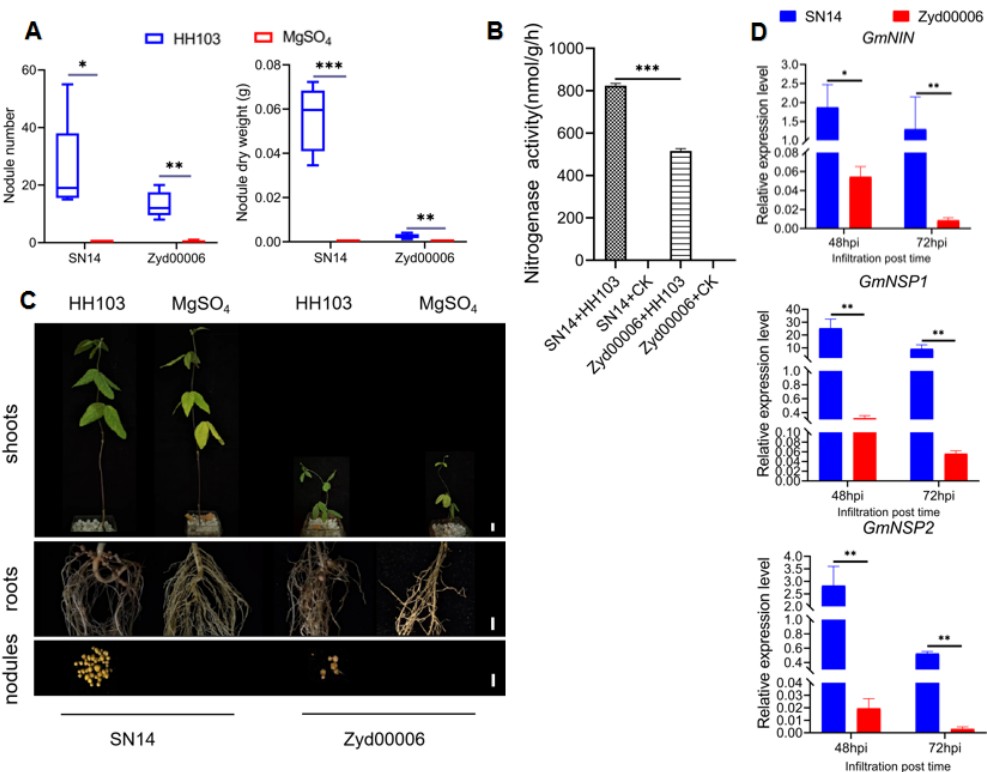

**Figure 1.** SN14 and Zyd00006 exhibited distinct phenotypic responses to *Sinorhizobium fredii* HH103. (**A**) Boxplots of NN and NDW. (**B**) After inoculation with *S. fredii* HH103, the nitrogenase activity of SN14 and Zyd00006 significantly varied. (**C**) Phenotypes of SN14 and Zyd00006 inoculated with *S. fredii* HH103 and MgSO4 buffer; shoot: scale bars represent 1.5 cm; root: scale bars represent 2 cm; nodule: scale bars represent 0.5 mm. (**D**) Expression of key nodulation-related genes in the early stage of rhizobial infection. The roots of SN14 and Zyd00006 were harvested at 3 dpi for assessing the relative transcript levels of *GmNIN*, *GmNSP1*, and *GmNSP2*. * indicates $p \leq 0.05$, ** indicates $p \leq 0.01$, *** indicates $p \leq 0.001$, based on Student's *t*-test.

**Table 1.** Parental and population statistics for nodule traits in the soybean 'SN14′ × 'Zyd00006′ population.

| | | CSSLs (*n* = 195) | | | Parents (Average) | |
|---|---|---|---|---|---|---|
| | **Traits** | **Average** | **Standard Deviation** | **Coefficient of Variation** | **SN14** | **Zyd00006** |
| HH103 | Nodule number | 48.38974 | 32.93076 | 0.68053 | 22.6 ± 55.3 | 12.6 ± 30.3 * |
| | Nodule dry weight (g) | 0.06418 | 0.03929 | 0.61216 | 0.02566 ± 0.000788 | 0.00974 ± 0.000012508 * |

\* indicates $p \leq 0.05$.

### 3.2. Identification of Genomic Regions Associated with Different Nodulation Phenotypes in Soybean CSSL Population

Due to differences in chromosome substitution fragments and wild soybean genome data, CSSLs exhibit distinct nodulation phenotypes. Therefore, we aimed to evaluate the potential nodulation-related phenotypic differences in the chromosomal regions in CSSLs. One QTL associated with NN, and two QTLs related to NDW were identified using genetic data and CSSL phenotypes after *S. fredii* HH103 infection. These QTLs exhibited LOD scores ranging from 2.6613 to 3.5025 and were found on chromosomes 02 (one QTL), 05 (one QTL), and 20 (one QTL) (Table 2). Block2959 was the substituted segment on chromosome 05, according to the whole-genome sequencing data, with an LOD of 3.2988. Block2959, which was 481 kb in size and contained 68 identified genes in accordance with the reference genome GlymaWm82.a2.v1 annotations, represented 8.8072% of the observed phenotypic variation (Figure 2; Table S3).

**Table 2.** Main QTLs identified in the CSSL population.

| QTL | Chr/LG | Start Position | End Position | LOD | PVE (%) | ADD |
|---|---|---|---|---|---|---|
| qNN-1 | Chr02/D1b | 43,852,453 | 43,880,682 | 3.5025 | 6.5895 | 32.6421 |
| qNDW-1 | Chr05/A1 | 41,425,769 | 41,907,158 | 3.2988 | 8.8072 | −0.0184 |
| qNDW-2 | Chr20/I | 47,302,953 | 47,897,802 | 2.6613 | 7.3791 | 0.0185 |

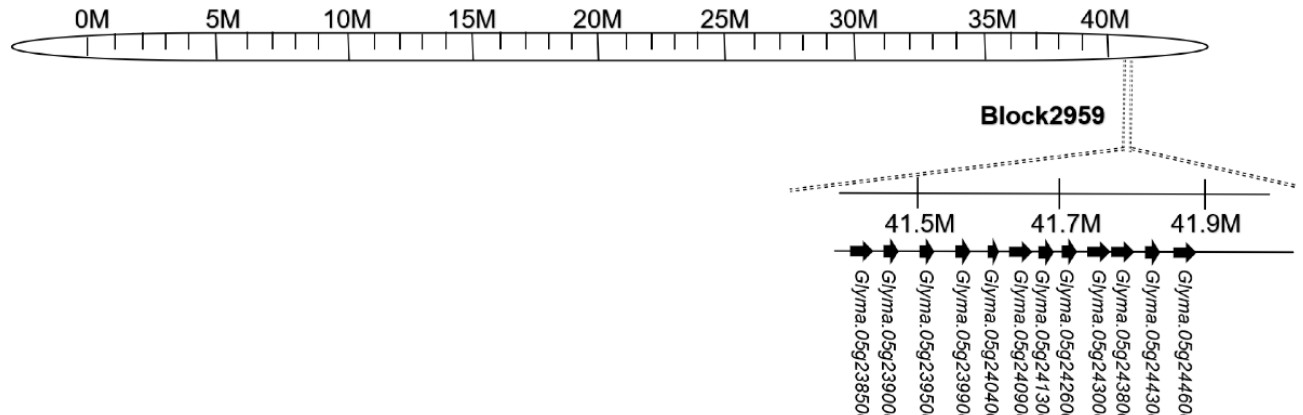

**Figure 2.** Physical map of the genomic areas connected to the QTL-determined nodule dry weight. The reference genome GlymaWm82.a2.v1 (https://jgi.doe.gov, retrieved on 3 May 2022) is the basis for the anticipated genes in the block.

### 3.3. Analysis of the Response of SN14 to S. fredii HH103 Infection Using RNA-Seq

To assess the variation in gene expression due to *S. fredii* HH103, RNA-seq was conducted to identify the candidate genes located in the QTLs. Root samples of SN14 were collected after independent inoculations with *S. fredii* HH103 and control treatment with MgSO₄ buffer. Pearson's correlation and principal component analysis of all RNA-seq libraries (Figure 3A) revealed that the biological repetition of this experiment was

strongly correlated. A Venn diagram was plotted, displaying the total number of genes with common differential expression across different samples. In total, 3663 DEGs were identified, with 1241 up- and 2422 downregulated DEGs (Figure 3B,C). This demonstrated that *S. fredii* HH103 infection significantly altered transcript profiles in soybean. Kyoto Encyclopedia of Genes and Genomes (KEGG) pathway enrichment analysis revealed that the upregulated DEGs were enriched in phenylpropanoid biosynthesis metabolism and environmental information processing for plant hormone signal transduction (Figure 3D). The downregulated DEGs were primarily enriched in photosynthesis-related pathways. Interestingly, the genes enriched in the plant–pathogen interaction pathway included both up- and downregulated DEGs, indicating that the immunological regulation of soybean after *S. fredii* HH 103 infection is complex (Figure 3E).

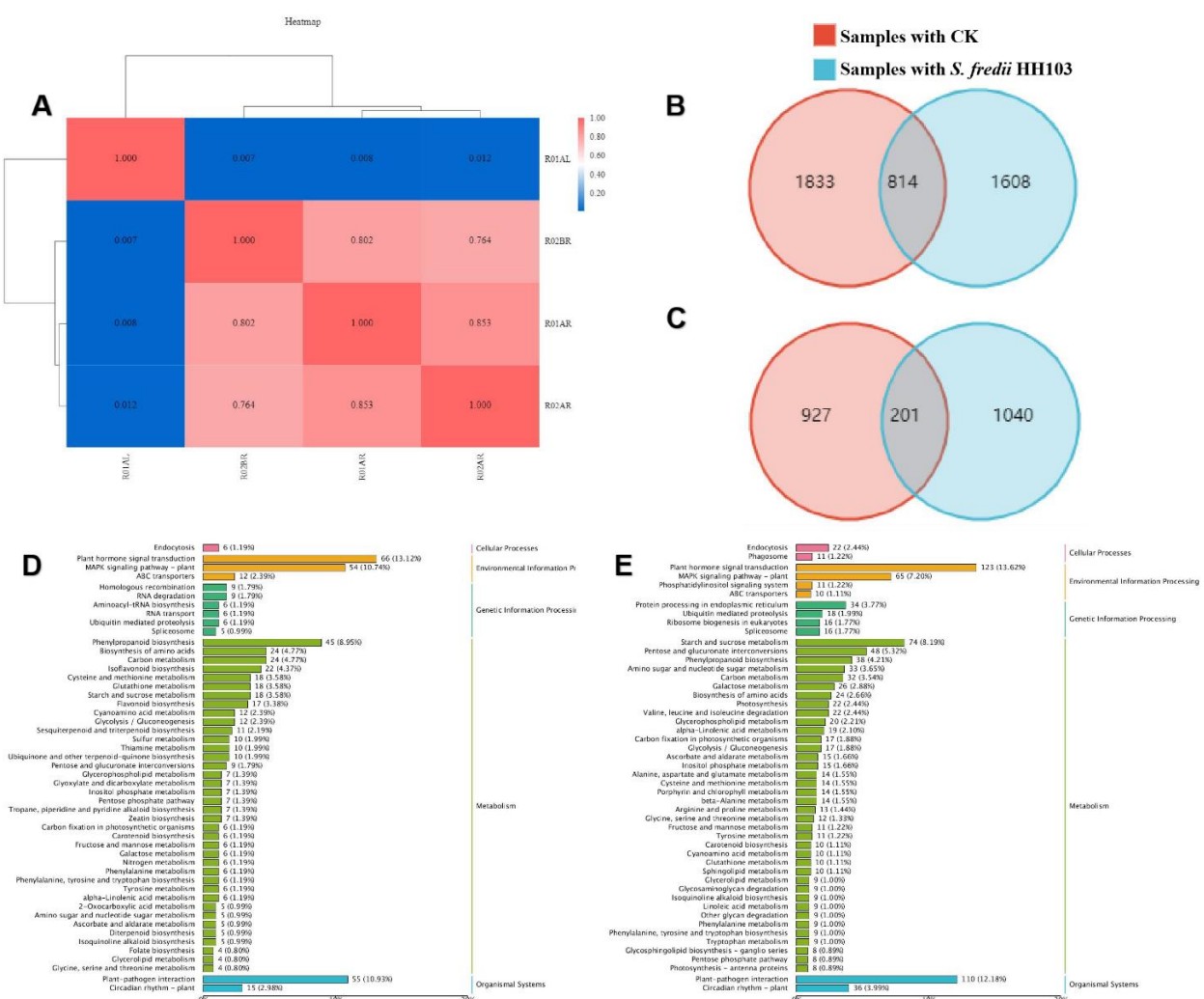

**Figure 3.** Transcriptome analysis of the roots of soybean inoculated with *S. fredii* HH103 and control treatment with MgSO$_4$ buffer (CK). (**A**) Expression correlation thermogram for all RNA-seq libraries. Venn diagram showing differentially expressed genes (DEGs) that were up- (**B**) and downregulated (**C**). Kyoto Encyclopedia of Genes and Genomes classifications of upregulated (**D**) and downregulated (**E**) DEGs.

### 3.4. Candidate Gene Prediction and qRT-PCR Validation

After combining the data from the QTL and the whole-genome sequencing analyses, this study focused on the genes located in the QTLs on chromosome 5. In the analysis of differentially expressed genes on chromosome 5 in conjunction with transcriptome sequencing, two DEGs (*Glyma.05g240500* and *Glyma.05g244200*) were found in the region

indicated by the QTL analysis and selected as prospective candidate genes for various nodulation phenotypes. QRT-PCR was performed to confirm the expression patterns of the two DEGs. The results were consistent with those of the transcriptome analysis. The expression of these two genes significantly increased at 12, 24, and 48 hpi with *S. fredii* HH103. However, *Glyma.05g240500* and *Glyma.05g244200* exhibited different expression patterns during 0–72 hpi with *S. fredii* HH103 (Figure 4). These results implied that the response of soybean to *S. fredii* HH103 infection may be mediated by these two candidate genes within the QTL.

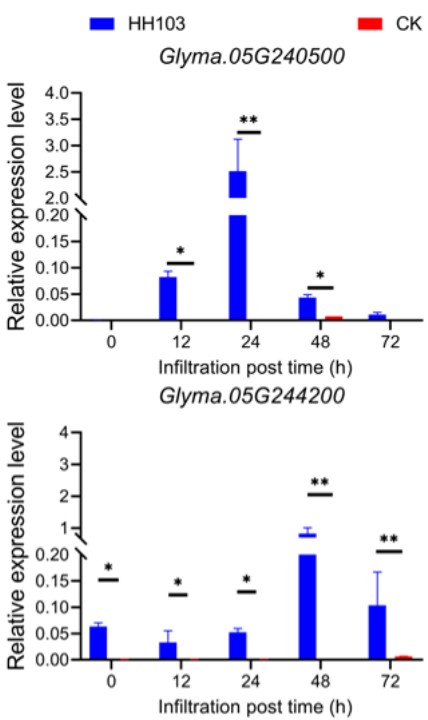

**Figure 4.** Expression profiling of candidate genes in SN14 in response to *S. fredii* HH103 infection. The y-axes indicate the relative expression levels between samples infected with *S. fredii* HH103 and control samples inoculated with MgSO4 buffer; the *x*-axes indicate the time points post-inoculation; * and ** indicate significant differences at $p < 0.05$ and $<0.01$, respectively, based on Student's two-tailed *t*-test.

*3.5. Glyma.05g240500 Promotes Nodulation in Transgenic Root Hairy Plants and Targeted Plant Nucleus and Cytosol*

*Glyma.05g240500,* involved in the root-specific co-expression sub-network, was selected for functional analysis. The 1077-bp *Glyma.05g240500* gene encodes a 359 amino-acid-long protein that belongs to the heat shock transcription factor (HSF) family and targets the plant nucleus and cytosol (Figure 5A). Phylogenetic analyses of 15 plants (including *G. max*, *G. soja*, *V. unguiculata*, *P. vulgaris*, *L. japonicus*, *C. arietinum*, *T. pratense*, *M. truncatula*, *A. hypogaea*, *L. albus*, *M. domestica*, *C. dentata*, *P. persica*, *P. trichocarpa*, and *T. cacao*) revealed that *Glyma.05G240500* was closely related to *GlysoPI483463.05G206500* and *Vigun03g010300* (Figure S1). The soybean transgenic hairy roots were constructed by transformation with *A. rhizogenes* strain K599 carrying pSoy1-*Glyma.05G240500*-GFP (for overexpression), pB7GWIWG2-*Glyma.05G240500*-DsRed (for RNA interference), and the corresponding empty vectors to further understand the function of *Glyma.05G240500* in soybean nodulation. And all the transgenic root hairy plants were inoculated with *S. fredii* HH103. After Glyma silencing, NN and NDW values were lower than those of the control hairy roots after *S. fredii* HH103 infection. The hairy roots overexpressing *Glyma.05G240500* exhibited higher NN and NDW values in comparison with the control hairy roots after *S. fredii* HH103 infection (Figure 5B,C). The hairy roots overexpressing *Glyma.05G240500*

exhibited higher SFW values in comparison with the control hairy roots and the hairy roots silencing *Glyma.05G240500* (Figure S2). These findings suggested that *Glyma.05G240500* may be involved in signal transduction that regulates nodulation when soybean recognizes *S. fredii* HH103.

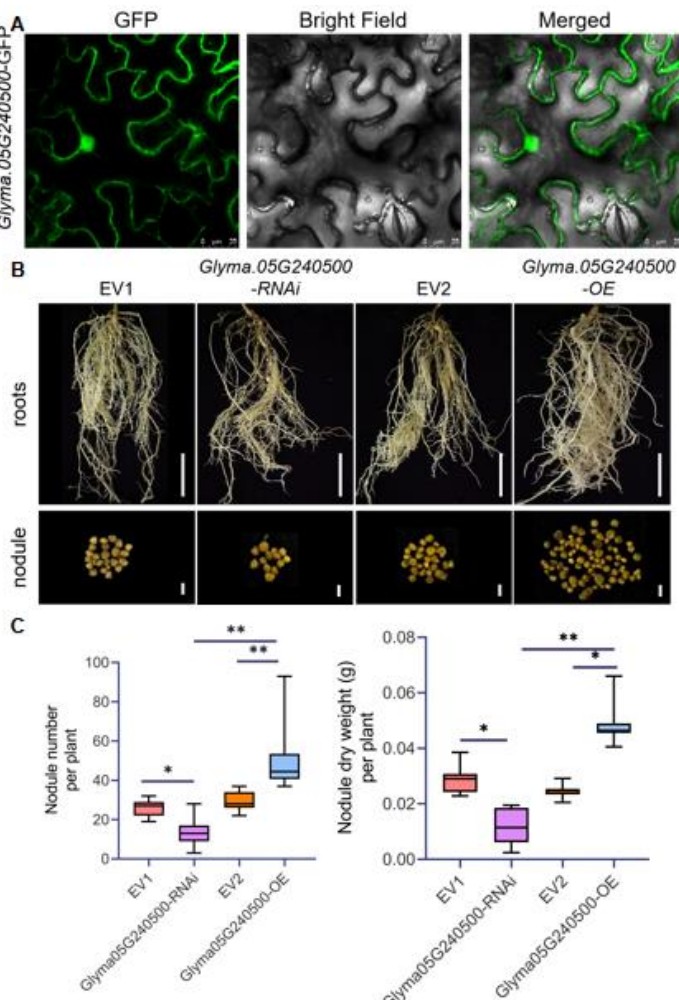

**Figure 5.** Glyma05G240500 targeted the plant nucleus and cytosol and promoted nodulation in transgenic hairy roots. (**A**) Analysis of subcellular localization of Glyma05G240500. (**B**) Nodular phenotypes of hairy roots transformed with EV1, *Glyma.05G240500*-RNAi, EV2, and *Glyma.05G240500*-OE after inoculation with HH103 (28 dpi). EV1, Empty vector for gene overexpression; root: scale bars represent 2 cm; nodule: scale bars represent 2 mm. (**C**) Boxplots of NN and NDW of hairy roots transformed with EV1, *Glyma.05G240500*-RNAi, EV2, and *Glyma.05G240500*-OE after inoculation with HH103 (28 dpi). * and ** indicate significant differences at $p < 0.05$ and $<0.01$, respectively, based on Student's two-tailed *t*-test.

### 3.6. Haplotype Analysis of Glyma.05g240500 in CSSL Population

Given that the CSSL population exhibited distinct phenotypic responses after *S. fredii* HH103 infection, the CSSL population and 685 natural varieties of soybean used in this study were sequenced to analyze *Glyma.05G240500* haplotypes (Haps) to confirm whether *Glyma.05G240500* could be related to *S. fredii* HH103. Three *Glyma.05G240500* Haps from 685 natural varieties of soybean and two *Glyma.05G240500* Haps from the CSSL population were identified, respectively, and two dominant Haps containing more than 10 accessions from the CSSL population and three dominant Haps from 685 natural varieties of soybean were identified. Besides, 4 SNPs each were identified within the promoter and CDS regions

of the CSSL population. Similarly, 13 and 6 SNPs, respectively, were identified within the promoter and CDS regions of the 685 natural varieties of soybean (Figure 6).

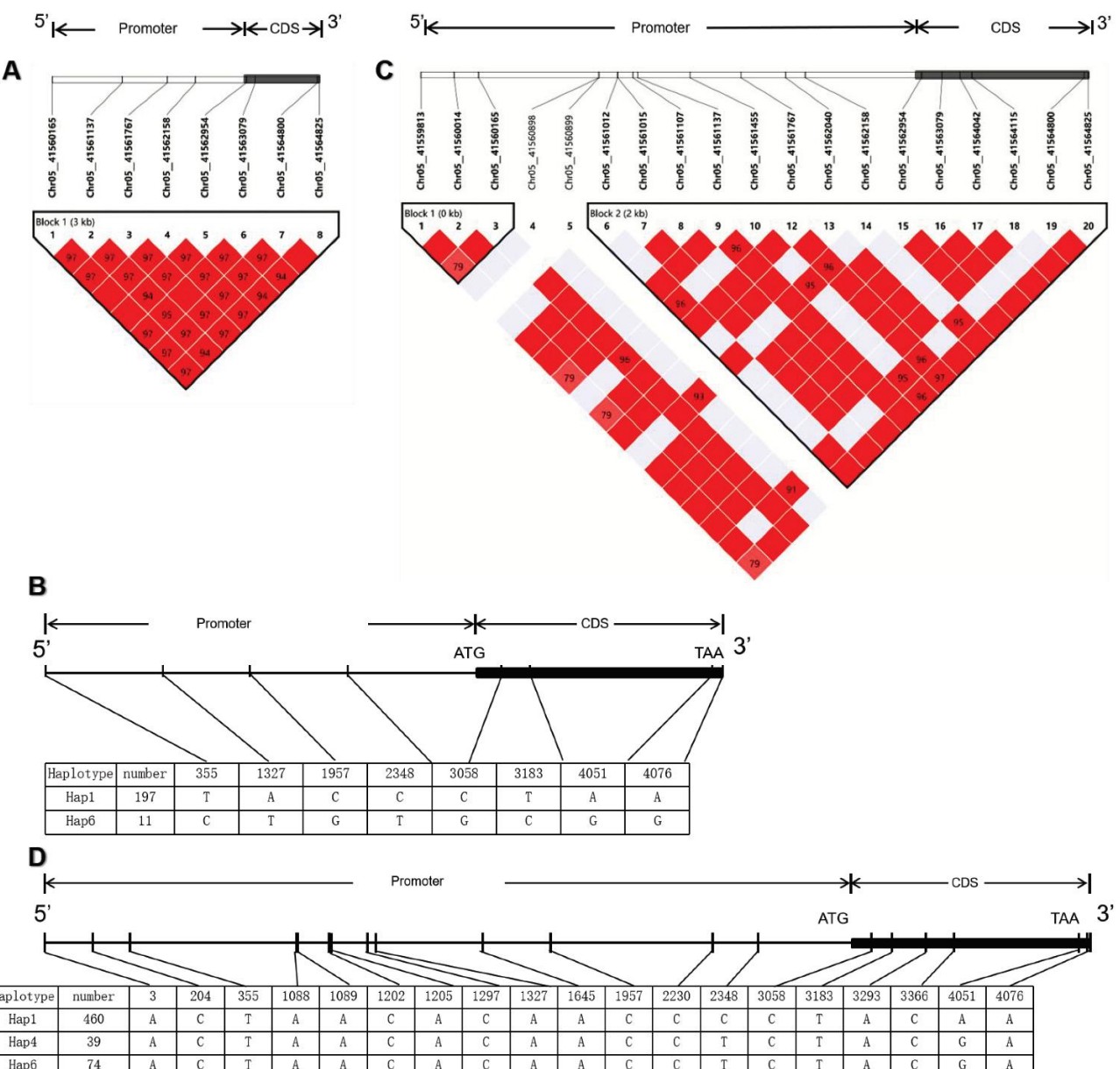

**Figure 6.** Haplotype analyses of Glyma05G240500. (**A**) Linkage disequilibrium analysis of SNPs in Glyma05G240500 from the CSSL population. (**B**) Haplotypes of Glyma05G240500 from the CSSL population. (**C**) Linkage disequilibrium analysis of SNPs in Glyma05G240500 from 685 natural varieties of soybean. (**D**) Haplotypes of Glyma05G240500 from 685 natural varieties of soybean. The color red of linkage disequilibrium analysis of SNPs indicates the degree of linkage between SNPs. Coding sequences and 3-kb sequences upstream of start codons were used for haplotype analyses.

## 4. Discussion

In this study, we identified two QTLs associated with the NDW on chromosomes Gm05 and Gm20, and a QTL related to the NN on chromosome Gm02. The locus on soybean chromosome Gm20 identified in this study aligns with a previously identified QTL associated with soybean maturity in the 5601T U99-310255 RIL population [34]. In our study, a QTL that had previously been linked to the oil content of soybeans was discovered to be adjacent to the QTL on Gm05 [35]. Additionally, a locus on Gm05 that was responsive

to NopT [19], a T3SS effector of *S. fredii* HH103, aligns with a QTL on Gm05 that was identified in this study.

Various novel putative genes engaged in several response pathways triggered by rhizobia infecting soybeans have been identified through the analysis of genetic populations created via a hybrid of cultivated and wild soybeans. NopD, the T3SS effector of *S. fredii* HH103, upgrades *S. fredii* HH103 infection by either directly or indirectly regulating *Glyma.19g068600* and *Glyma.19g069200* expression. This occurs when rhizobia and soybean plants form a symbiotic relationship [20]. The CSSL population used in this study was produced via multiple recurrent crossings. The nodulation phenotypes of the CSSL population varied when infected with *S. fredii* HH103, which could be attributed to the substituted segments affecting signaling pathways involved in soybean nodulation. Gene annotation, information on the parental genomes, and qPCR analyses of the QTLs were combined to determine the relationship between *Glyma.05g240500* and HH103. A 359-amino-acid protein from the soybean HSF family is encoded by *Glyma.05g240500*. The regulatory network for the response of plants to heat stress mainly consists of HSFs. They participate in multiple processes regulating transcription and are crucial for the signaling of heat stress and responses to several other abiotic stresses [36]. HSF family members have been identified in various plant species, such as soybean, moso bamboo, tomato, lettuce, *Arabidopsis*, and rice [37–41]. HSFs play a wide range of roles in stress resistance, including defense against protein misfolding, inflammatory reactions, and environmental stressors [42]. Cis-elements found in the promoter regions of HSFs in soybean are probably involved in the responses to ABA, low temperatures, and drought stress. Moreover, under stress conditions, the transcriptional levels of HSFs exhibited significant variation compared with the transcriptional levels under no stress [37]. MdHSFA8a, a HSF in *M. domestica*, can promote flavonoid accumulation under drought stress [43]. HSF encoded by *Glyma.05g240500* shared an 80% similarity with MdHSFA8a (Figure S1). From the nodulation tests in this study, hairy roots overexpressing *Glyma.05G240500* exhibited higher NN and NDW, suggesting that *Glyma.05G240500* may positively impact the development of nodules and rhizobia infection. In addition, *Glyma.05G240500* had a potential regulatory role in the expression of genes related to tocopherol biosynthesis [44]. Glyma.05g240500 was introduced into the STRING database to construct the protein–protein interaction network (Figure 7). The color of the protein represents its importance in the interaction network. The redder the color, the more important it is in the interaction network. Lines and arrows between proteins represent their interactions. Further, putative interaction partners of Glyma.05G240500 were identified via the STRING database. Glyma.05G34450 was annotated as a HSF_DOMAIN domain-containing protein, as the key protein is reported to interact with 10 other proteins in the protein–protein interaction network. *Glyma.05G34450* annotation version Wm82.a1.v1.1 and *Glyma.05g240500* annotation version Wm82.a2.v1 correspond to the same gene. The GO annotation of Glyma.05G34450 revealed that the gene was associated with positive regulation of the metabolic processes of nitrogen compounds. A protein interaction network containing the HSF_DOMAIN domain-containing protein may provide an explanation or hypothesis for further study on how *S. fredii* HH103 regulates the symbiotic nitrogen fixation signal pathway.

Different soybean germplasms have altitudinal genetic diversity with the potential to improve soybean breeding and economic characteristics. Compared with the improved germplasm, high levels of genotypic and phenotypic variations can be found in wild soybeans [45,46]. Wild soybeans can acquire biological stress tolerance and abiotic stress adaptation through natural selection without the use of artificial selection [47,48]. In this study, approximately 95% of the genomic information of wild soybean Zyd00006 was covered by the substituted wild soybean genomic segments. However, new candidate genes still need to be accurately located among the remaining large and tiny fragments of the wild soybean genome. In this study, we revealed *Glyma.05G240500* contributing to symbiosis with *S. fredii* HH103, which provided a reference for using SNF and marker-assisted selection in molecular breeding. Combined with the results of analyzing the

haplotypes of *Glyma.05G240500* in this study, potential high quality breeding materials could be obtained, and the screened materials could be used to validate the efficiency and accuracy of molecular markers for molecular-assisted selection. Haplotype combinations with optimal symbiotic nodulation and other agronomic traits could be screened. Propose supporting algorithms to discover breeding material selection strategies and molecular breeding strategies for different combinations of traits such as high nodulation and high yield, high nodulation and resistance, etc.

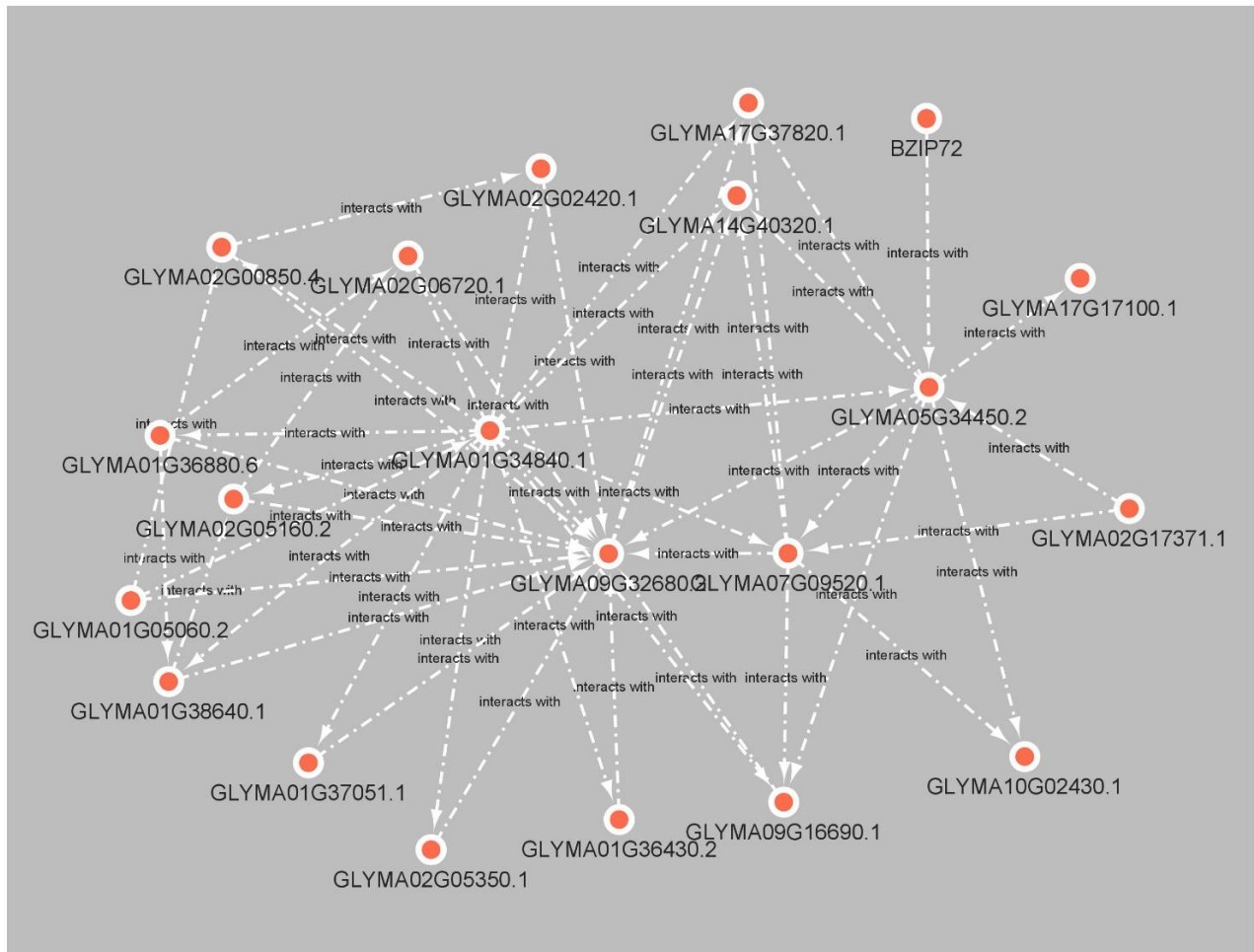

**Figure 7.** Protein–protein interaction network of candidate genes in soybean established through STRING analysis.

## 5. Conclusions

In this study, the chromosome region in soybean responsive to *S. fredii* HH103 was identified using a CSSL population with a genomic background of wild soybean. *Glyma.05G240500* responding to *S. fredii* HH103 was identified through gene annotation, parental genomic information, and qRT-PCR analyses of the QTLs. *Glyma.05G240500* encodes a 359-amino-acid protein belonging to the HSF family of soybeans. Glyma.05G240500 targeted the plant nucleus and cytosol and positively regulated nodulation in soybean. Our study revealed a soybean gene contributing to symbiosis with *S. fredii* HH103 and provided a reference for using SNF and marker-assisted selection in molecular breeding.

**Supplementary Materials:** The following supporting information can be downloaded at: https: //www.mdpi.com/article/10.3390/agronomy13082037/s1, Table S1: Information on Strains and Vectors [49–52]; Table S2: Primers for this research; Table S3: Functional annotation of candidate genes identified in genomic regions; Figure S1: Phylogenetic analysis of the candidate gene using the corresponding gene from different plants (including *Glycine max*, *Glycine soja*, *Vigna unguiculata*, *Phaseolus vulgaris*, *Lotus japonicus*, *Cicer arietinum*, *Trifolium pratense*, *Medicago truncatula*, *Arachis hypogaea*, *Lupinus albus*, *Malus domestica*, *Castanea dentata*, *Prunus persica*, *Populus trichocarpa*, and *Theobroma cacao*); Figure S2: Boxplots of SFW of hairy roots transformed with EV1, *Glyma.05G240500*-RNAi, EV2 and *Glyma.05G240500*-OE after inoculation with HH103 (28 dpi).

**Author Contributions:** Conceptualization, D.X. and J.W.; methodology and figure creation, H.N. and S.T.; software and data analysis, G.Z. and H.T.; validation, Y.P. and K.Y.; formal analysis, H.N.; investigation, C.L. and Q.C.; resources, J.H. and J.W.; data curation, S.T.; writing—original draft preparation, H.N. and S.T.; writing—review and editing, D.X. and C.L.; visualization, H.N. and H.T.; supervision, Q.C.; project administration, C.L.; funding acquisition, C.L., D.X. and H.N., H.N. and S.T. contributed equally to this work. All authors have read and agreed to the published version of the manuscript.

**Funding:** This research was funded by the China Postdoctoral Science Foundation (2023MD734142), the Postdoctoral Foundation of Heilongjiang Province (LBH-Z22080), and the National Natural Science Foundation of China (Grant numbers: 32072014, 32201809, 32272072, and U20A2027).

**Data Availability Statement:** Not applicable.

**Conflicts of Interest:** The authors declare no conflict of interest.

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
