# Peer review of "QTL Mapping and Functional Identification of Candidate Genes Regulated by Sinorhizobium fredii HH103 and Associated with Nodulation Traits in Soybean"

_agronomy, doi:10.3390/agronomy13082037_

Round 1
Reviewer 1 Report
This manuscript presents a large amount of data that represents a lot of work. It identifies 3 QTLs that may contribute to nodule function and then looks at candidates within one to try and identify the causative gene/s. However the results of the gene characterization are not particularly convincing. Real-time PCR is used to look at expression of two genes and because this changes after inoculation the authors state "These results indicated that the response of soybean 274 to S. fredii HH103 infection is mediated by these two candidate genes within the QTL." There is no comparison with other genes in the QTL. One gene is then silenced and overexpressed and this changes nodule number and nodule dry weight but there is no analysis of how this affects nitrogen fixation. Although hairy root transformed plants may not be a perfect way to assess overall nitrogen fixation a comparison of the shoots of the plants could give an indication as to whether the change in nodule number increases nitrogen fixation and shoot growth.
The characterized gene is a transcription factor but there is no characterization of its targets to point to how it might regulate nodule number or nodule dry weight. Without this no real conclusions can be made about the role of this gene or its importance.
Author Response
Dear Editors and Reviewers:
Thank you for the referee’s comments concerning our manuscript entitled‘QTL Mapping and Functional Identification of Candidate Genes Regulated by Sinorhizobium fredii HH103 and Associated with Nodulation Traits in Soybean’. We have studied their comments carefully and have made correction which we hope meet with their approval. We hope that the revision is acceptable and look forward to hearing from you soon. Thank you and best regards.
Yours sincerely,
Dawei Xin and Chunyan Liu
Answer to reviewer1:
- Line274: the results of the gene characterization are not particularly convincing. Real-time PCR is used to look at expression of two genes and because this changes after inoculation the authors state "These results indicated that the response of soybean to S. fredii HH103 infection is mediated by these two candidate genes within the QTL." There is no comparison with other genes in the QTL.
Answer: Thank you for your suggestion, we have revised the manuscript as your suggestion, as shown in red font. After combining the data of the QTL and the whole-genome sequencing analyses, this study focused on the genes located in the QTLs on chromosome 5. Analysis of differentially expressed genes on chromosome 5 in conjunction with transcriptome sequencing, two DEGs (Glyma.05g240500 and Glyma.05g244200) were found in the region indicated by the QTL analysis and selected as prospective candidate genes for various nodulation phenotypes.
In the follow-up study, we will carry out stable genetic transformation and gene function analysis of the candidate gene to clarify the deep-rooted mechanism of the gene regulating symbiotic nitrogen fixation. Meanwhile, we will also design experiments to verify whether other genes within this QTL interval affect symbiotic nodulation.
- One gene is then silencedand overexpressed and this changes nodule number and nodule dry weight but there is no analysis of how this affects nitrogen fixation.
Answer: Thank you for your suggestion, we added the relationship between soybean nodule formation and nitrogen fixation in introduction. Increasing nodule formation enhanced soybean utilization of N and P, which enhanced yield performance, and symbiotic nitrogen fixation (SNF) ability of leguminous plants is considered a complex quantitative trait, involving many related characteristics such as nodule number (NN), nodule dry weight (NDW), etc.
And I discussed in the Discussion section, from lines 373 to lines 391, the progress of researches on Glyma.05g240500 in other crops and the function of the proteins encoded by homologous genes of Glyma.05g240500, in an attempt to explain the effect of silencing or overexpression of Glyma.05g240500 on the symbiotic nodule phenotype. As shown in red font.
lines 373 to lines 391:Gene annotation, information of the parental genomes, and qPCR analyses of the QTLs were combined to determine the relationship between Glyma.05g240500 and HH103. A 359-amino-acid protein from the soybean HSF family is encoded by Glyma.05g240500. The regulatory network for the response of plants to heat stress mainly consists of HSFs. They participate in multiple processes regulating transcription and are crucial for the signaling of heat stress and responses to several other abiotic stress[36]. HSF family members have been identified in various plant species, such as soybean, moso bamboo, tomato, lettuce, Arabidopsis, and rice[37-41]. HSFs play a wide range of roles in stress resistance, including defense against protein misfolding, inflammatory reactions, and environmental stressors[42]. Cis-elements found in the promoter regions of HSFs in soybean are probably involved in the responses to ABA, low temperature, and drought stress. Moreover, under the stress conditions, the transcriptional levels of HSFs exhibited significant variation compared with the transcriptional levels under no stress[37]. MdHSFA8a, a HSF in M. domestica, can promote flavonoid accumulation under drought stress[43]. HSF encoded by Glyma.05g240500 shared 80% similarity with MdHSFA8a (Figure S1). From the nodulation tests in this study, hairy roots overexpressing Glyma.05G240500 exhibited higher NN and NDW, suggesting that Glyma.05G240500 may positively impact the development of nodules and rhizobia infection.
And I analysed the annotation of the signalling pathways involved in Glyma.05G240500 in the discussion from lines 401 to lines 403. As shown in red font.
lines 401 to lines 403:Glyma.05G34450 annotation version is Wm82.a1.v1.1 and Glyma.05g240500 annotation version Wm82.a2.v1 correspond to the same gene. GO annotation of Glyma.05G34450 revealed that the gene was associated with positive regulation of metabolic process of nitrogen compounds.
- Although hairy root transformed plants may not be a perfect way to assess overall nitrogen fixation a comparison of the shoots of the plants could give an indication as to whether the change in nodule number increases nitrogen fixation and shoot growth.
Answer: Thank you for your suggestion. In this study, we collected the shoot fresh weight (SFW) of transgenic hairy root plants, but it was not shown in the results during the previous writing. We added the result in the supplemental figures. The hairy roots overexpressing Glyma.05G240500 exhibited higher SFW values in comparison with the control hairy roots and the hairy roots silencing Glyma.05G240500.
- The characterized gene is a transcription factor but there is no characterization of its targets to point to how it might regulate nodule number or nodule dry weight. Without this no real conclusions can be made about the role of this gene or its importance.
Answer: Thank you for your suggestion. I annotated and analysed the signalling pathways involved in Glyma.05G240500 in the discussion. And in the discussion section, I discussed the progress of researches on Glyma.05G240500 in other crops, as well as analysing the proteins encoded by the homologues of Glyma.05G240500, and analysing the reasons why this gene affects the nodulation phenotypes NN and NDW.
At present, we are screening the targets of transcription factor, but there is graduation pressure at present, so the further research on the targets of transcription factor can not be shown in this study. At present, we have predicted the target genes of this transcription factor on JASPAR website which is a bioinformatics site for predicting transcription factor target genes, and are currently constructing the related vectors of the target genes, so as to verify the interaction between the transcription factor and the promoter element of the target genes in the later stage.

Reviewer 2 Report

The English writing in the manuscript is generally good. However, there are some areas where improvements can be made, primarily focusing on inconsistencies in denotation throughout the manuscript. It is recommended to carefully review the manuscript for any inconsistencies in terminology, abbreviations, and formatting to ensure clarity and maintain a consistent writing style. Additionally, proofreading the manuscript to identify and correct any typographical errors will further enhance the overall quality of the writing.
Author Response
Dear Editors and Reviewers:
Thank you for the referee’s comments concerning our manuscript entitled‘QTL Mapping and Functional Identification of Candidate Genes Regulated by Sinorhizobium fredii HH103 and Associated with Nodulation Traits in Soybean’. We have studied their comments carefully and have made correction which we hope meet with their approval. We hope that the revision is acceptable and look forward to hearing from you soon. Thank you and best regards.
Yours sincerely,
Dawei Xin and Chunyan Liu
Answer to reviewer2:
- Line 29: mentions “role of a soybean gene”: what gene this this refer to?
Answer: Thank you for your suggestion. The gene mentioned in Line 29 was referred to Glyma.05G240500. We have revised the abstract section to further improve the quality,as shown in red words.
- “provides a reference for facilitating symbiotic nitrogen fixation and marker-assisted selection” This is reiterated in line 386 of the conclusion. However, the manuscript would benefit from providing a clearer explanation of how the authors envision the practical implementation of marker-assisted breeding using the findings presented. It is important to consider that not all readers may be familiar with marker-assisted breeding techniques.
Answer: Thank you for your suggestion. We added this section in discussion, as shown in red font. Combined with the results of analyzing the haplotypes of Glyma.05G240500 in this study, potential high quality breeding materials could be obtained and the screened materials could be used to validate the efficiency and accuracy of molecular markers for molecular assisted selection. Haplotype combinations with optimal symbiotic nodulation and other agronomic traits could be screened. Propose supporting algorithms to discover breeding material selection strategies and molecular breeding strategies for different combinations of traits such as high nodulation and high yield, high nodulation and resistance, etc.
- Line 41: “ and even atmosphere” : The manuscript mentions the "deterioration condition ofthe atmosphere" without explicitly clarifying its meaning. It is unclear whether this refers to the deterioration caused by nitrogen deposition dispersed through the air. Providing a more precise explanation of the term would help readers better understand the specific environmental concerns associated with the overuse of nitrogen fertilizers and the relevance of biological nitrogen fixation as an environmentally friendly alternative.
Answer: Thank you for your suggestion. We have revised the manuscript as your suggestion, as shown in red font. The overuse of nitrogen fertilizers has resulted in significant environmental pollution, deteriorating the quality of the soil, water resources, and even atmosphere due to the fact that a small amount of nitrous oxide released during nitrification mixes into the stratosphere, where it destroys ozone.
- Line 50: The sentence, "100-seed weight (100-SW), a key objective characteristic in field breeding, is a significant contributor of soybean production," could benefit from further clarification regarding the specific reasons why 100-SW is considered a significant contributor to soybean yield. While it may be inferred that the seeds are the consumed parts of the soybean plant, it would be helpful to explicitly mention this fact. Additionally, for readers who are not familiar with the soybean field, it would be beneficial to provide a brief explanation of why 100-SW is an important trait for soybean production.
Answer: Thank you for your suggestion, we added the statements as your suggestion, as shown in red font. 100-seed weight (100-SW)is an important component of soybean yield traits and can characterise soybean yield to a certain extent.
- Line 60: "Three and five quantitative trait loci (QTLs) for the percentage of nitrogen derived," is unclear in terms of the number of QTLs identified. It is unclear whether the authors are referring to three QTLs or five QTLs. To avoid confusion, it would be beneficial for the authors to provide a more precise statement regarding the number of QTLs identified for the percentage of nitrogen derived. Additionally, it would be helpful to provide a brief explanation of what the percentage of nitrogen derived refers to, particularly for readers who are not familiar with this specific trait in the context of soybean research
Answer: Thank you for your suggestion. We added more precise statement regarding the number of QTLs identified for the percentage of nitrogen derived and explanation about the percentage of nitrogen derived as your suggestion, as shown in red words. Total nitrogen derived from atmosphere (Ndfa) refers to the amount of nitrogen fixed from the atmosphere through symbiotic nitrogen fxation. Three QTLs for percentage of nitrogen derived from the atmosphere (%Ndfa) and five QTLs for Ndfa were identified respectively.
- Line 82: "a population of chromosomal segment substitution lines (CSSLs) was produced" lacks clarity regarding whether these lines were previously generated in another study or as part of the current study. Please provide a clear explanation of whether the CSSL population was specifically created for this research or if it was previously established.
Answer: Thank you for your suggestion. We added more precise statement regarding the CSSL population in Line 92 as your suggestion. The CSSL population was previously established according to the 22th reference.
- Line 88: “Glyma.05G240500” was previously already picked up in a QTL analysis in a study by Zhang et al.. Yet this paper is not cited or discussed.
Answer: Thank you for your suggestion. We added discussion regarding “Glyma.05G240500” in Line 373, and the study by Zhang et al. was added to reference as your suggestion.
- Line 110: there seems to be an additional space between “26 °C” while in other parts of theMaterials and methods this space is not present (e.g. line 95). Please be concise throughout the manuscript. I recommend removing the additional space to ensure consistency with the formatting used in other parts of the Materials and Methods section.
Answer: Thank you for your suggestion, we revised the manuscript as your suggestion.
- Section 2.3, lines 116-123, lacks clarity regarding the sequencing of the 685 soybean wild types. It is crucial to provide a clear explanation of whether the wild types were sequenced as part of the current study or if they were sequenced in a previous study. If the wild types were sequenced as part of this study, I recommend depositing the sequenced data in a public repository to ensure transparency and accessibility. Furthermore, if the wild types were included in a previous study, it is essential to refer to that study explicitly and provide clear attribution.
Answer: Thank you for your suggestion. We have revised the manuscript as your suggestion. The 685 soybean wild types were sequenced for the current study, but the sequenced data is unpublished. We have a total of 1,000 germplasm resources that have had their genomes sequenced, but only 685 sequencing results are available at this time. We originally planned to use the genome sequencing results of 1000 germplasm resources to dig deeper into the genetic loci of multiple traits, so it is not convenient to upload the data at present.
- Line 120: “FastP was used to ensure that the raw data was of high quality. After collecting clean data”: This implies that some filtering for low quality was done. Please specify the quality filtering parameters applied using FastP or another filtering method and mention the version used.
Answer: Thank you for your suggestion, we added the quality filtering parameters applied using FastP and the version used as your suggestion, as shown in red font. The quality filtering parameters applied using FastP was “-c-f10-F10”,with the version 0.23.1.
- Line 121: "the Williams 82 assembly V2.0 was mapped using the Burrows-Wheeler Alignment tool," may lead to confusion as it implies that the assembly itself was being mapped. Also here please include a version that was used.
Answer: Thank you for your suggestion.We added more precise statement as your suggestion, as shown in red words. Williams82 v2.0 was used as the reference genome, and the Burrows-WheelelAlignment (BWA, Version: 0.7.17-r1188) tool was used for mapping.
- Line 127: "Due to variations in chromosomal substituted regions and genetic information from wild soybean, the nodulation phenotypes of the CSSL population inoculated with HH103 were different," lacks clarity and does not provide a clear explanation of how the data was used to obtain the results.
Answer: Thank you for your suggestion.We added more precise statement as your suggestion. Due to variations in chromosomal substituted regions and genetic information from wild soybean, the nodulation phenotypes, such as NN, NDW and nitrogenase activity, of CSSL population inoculated with HH103 were different.
- Section 2.8: The title "Phylogenetic analysis of Candidate Genes" implies a plural analysis, while it is unclear from the context if the analysis was conducted for multiple genes or just a single gene. Furthermore, it is important to clarify how the orthologs of the gene(s) were identified.
Answer: Thank you for your suggestion.We revised the statement as your suggestion.
- Line 193: “resequenced” The term "resequenced" requires clarification to provide a better understanding of how the population was resequenced, which is not mentioned in this section.
Answer: Thank you for your suggestion, we revised the manuscript as your suggestion. The term "resequenced" means “whole-genome sequencing”.
- Figure 1: The labels of the plots are currently ordered as A, D, B. To maintain alphabetical order, I recommend renaming the plots as A, B, D. Additionally, the bottom plot of D is missing an x-label. Please ensure that the x-axis is labeled appropriately for the plot in question. Furthermore, the error whiskers in the plot for D either appear disrupted or are missing. Please revise the plot to ensure that the error whiskers are properly displayed, providing a clear representation of the data. It is also worth considering adding labels or symbols to indicate the significance between the phenotypes, if applicable similar to what is done in figure 4. Lastly, in line 220, there appears to be a missing space between "NDW." and "(B)".
Answer: Thank you for your suggestion. We revised the Figure 1 as your suggestion.
- Line 228: unclear sentence what was revealed and how?
Answer: Thank you for your suggestion. We have revised the section to further improve the quality.
- Figure 2/ Table2: In line 241, the phrase "phenotypes of nodules" appears, but based on the text, it is implied that this refers only to NDW (nodule dry weight). Regarding the three QTLs mentioned in Table 2, it is crucial to address why the authors chose to focus only on the QTL found on chromosome 5 and neglected the other QTLs for further analysis. It would be beneficial to provide an explanation or rationale for this decision. It is also worth noting that the supplemental table S3 only contains data on the QTL of chromosome 5. To ensure completeness and transparency, it would be advantageous to either include data on the other QTLs or clarify in the manuscript why those QTLs were not included in the supplemental table.
Answer: Thank you for your suggestion, we revised the manuscript as your suggestion. The reason for focusing only on the QTL found on chromosome 5 is that the QTL found on chromosome 5 overlapped with the Block 2959 of chromosome substitution fragments according to the genome sequencing data, as was not found on another two chromosomes.
- Figure 3/4: The samples labeled as "CK" are mentioned in the figure but are not explicitly explained in the text.
Answer: Thank you for your suggestion.We added more precise statement as your suggestion, as shown in red words. Figure 3. Transcriptome analysis of the roots of soybean inoculated with S. fredii HH103 and control treatment with MgSO4 buffer (CK).
- Figure 4: “using SPSS 12.0” should not be included in the figure caption but in the materials and methods. And again the x-label is missing on the bottom plot in figure 4.
Answer: Thank you for your suggestion. We revised the Figure 4 and figure caption as your suggestion.
Figure 4. Expression profiling of candidate genes in SN14 in response to S. fredii HH103 infection. The y-axes indicate the relative expression levels between samples infected with S. fredii HH103 and control samples inoculated with MgSO4 buffer; the x-axes indicate the time points post-inoculation; * and ** indicate significant differences at p < 0.05 and < 0.01, respectively, based on Student’s two-tailed t-test.
- Line 295: It seems there is a discrepancy between the repeated mention of "NDW" (nodule dry weight) in the text and the absence of specific weight observations. Please revise the manuscript by including the actual nodule dry weight observations in the relevant sections.
Answer: Thank you for your suggestion, the actual nodule dry weight can be observed in Figure 5C.
- Figure 5 B: I believe that the phylogenetic tree included in the figure does not contribute significantly to the overall message and can be better presented in a supplemental file, consider moving the phylogenetic tree to a supplementary figure or file. This will allow readers who are specifically interested in the phylogenetic analysis to access and review it separately and be able to read the labels. By doing so, the main figure can focus on the key findings and results without unnecessary complexity.
Answer: Thank you for your suggestion, we revised the manuscript as your suggestion.
- Line 315: Please explain what is meant by "resistance to fredii HH103" in the context of the study. Consider revising the sentence to provide a brief explanation of what this resistance entails, whether it refers to a reduced nodulation response, limited colonization, or any other specific characteristics observed in the interaction between soybean and S. frediiHH103. Regarding the statement "Three and two Glyma.05G240500 Haps were identified," it is unclear what is being referred to.
Answer: Thank you for your suggestion.The word “resistance” was used inappropriately in the original article, causing ambiguity in the reading, and we have revised this section, as shown in red words. to confirm whether Glyma.05G240500 could be related to S. fredii HH103.
Regarding the statement "Three and two Glyma.05G240500 Haps were identified," ,We added more precise statement as your suggestion, as shown in red words. Three Glyma.05G240500 Haps from 685 natural varieties of soybean and two Glyma.05G240500 Haps from the CSSL population were identified, respectively, and two dominant Haps containing more than 10 accessions from the CSSL population and three dominant Haps from 685 natural varieties of soybean were identified.
- Figure 6 Line 321: the current figure caption is short and does not provide a clear explanation of the figure. Please revise the figure caption to include a more detailed description of the figure components such as the meaning of the red boxes, and the tables under B and D.
Answer: Thank you for your suggestion.We added more precise statement as your suggestion, as shown in red words.
Figure 6. Haplotype analyses of Glyma05G240500. (A and C) Linkage disequilibrium analysis of SNPs in Glyma05G240500 from the CSSL population and 685 natural varieties of soybean, respectively. Color from blue to red indicates the degree of linkage between SNPs. (B and D)
- Line 344: "the relationship between Glyma.05g240500 and HH103," it is important to convey a more definitive tone in the manuscript after the observation has been presented. Consider revising the sentence to reflect a clearer understanding or interpretation of the relationship.
Answer: Thank you for your suggestion, we revised the manuscript as your suggestion.
- Line 361/Figure 7: The figure is mentioned here in the text but it lacks further explanation, it is crucial to provide a more detailed description of the figure, its purpose, and the key findings it presents. Additionally, if the analysis using STRING is not explained in the Materials and Methods section. The labels in the figure overlap each other and are hard to read. Overall, it remains unclear what Figure 7 adds to the manuscript.
Answer: Thank you for your suggestion.We added more precise statement as your suggestion.
- The supplemental figures in the manuscript only contain a limited number of tables and do not provide comprehensive additional data. Notably, there is no information on the differential expression analysis or the candidates found on other QTLs. Furthermore, there is no Data Availability section in the manuscript with details on the deposition or public availability of the sequenced data presented in this manuscript.
Answer: Thank you for your suggestion. We focus on the QTL found on chromosome 5 in this work because chromosome 5 overlapped with the Block 2959 of chromosome substitution fragments according to the genome sequencing data, as was not found on chromosome 2 and chromosome 20. We have a total of 1,000 germplasm resources that have had their genomes sequenced, but only 685 sequencing results are available at this time. We originally planned to use the genome sequencing results of 1000 germplasm resources to dig deeper into the genetic loci of multiple traits, so it is not convenient to upload the data at present.
- The author contributions provided in the manuscript are vague and do not clearly indicate the specific responsibilities of each author in terms of figure creation or data analysis. Please provide more detailed and specific information about the contributions of each author, particularly in relation to figure creation and data analysis.
Answer: Thank you for your suggestion. We added more precise statement as your suggestion.
